# Resveratrol Microencapsulation into Electrosprayed Polymeric Carriers for the Treatment of Chronic, Non-Healing Wounds

**DOI:** 10.3390/pharmaceutics14040853

**Published:** 2022-04-13

**Authors:** Andrea De Pieri, Keegan Ocorr, Kyle Jerreld, Mikkael Lamoca, Wolfgang Hitzl, Karin Wuertz-Kozak

**Affiliations:** 1Department of Biomedical Engineering, Rochester Institute of Technology (RIT), 106 Lomb Memorial Rd., Rochester, NY 14623, USA; adpbme@rit.edu (A.D.P.); kjo1193@g.rit.edu (K.O.); kpj8031@rit.edu (K.J.); mil9630@g.rit.edu (M.L.); 2Research and Innovation Management (RIM), Biostatistics and Publication of Clinical Trial Studies, Paracelsus Medical University, 5020 Salzburg, Austria; wolfgang.hitzl@pmu.ac.at; 3Department of Ophthalmology and Optometry, Paracelsus Medical University, 5020 Salzburg, Austria; 4Research Program Experimental Ophthalmology and Glaucoma Research, Paracelsus Medical University, 5020 Salzburg, Austria; 5Schön Clinic Munich Harlaching, Spine Center, Academic Teaching Hospital and Spine Research Institute of the Paracelsus Medical University Salzburg (Austria), 81547 Munich, Germany

**Keywords:** electrospraying, microparticles, drug delivery, inflammation, matrix metalloproteinases (MMPs), extracellular matrix (ECM) deposition

## Abstract

Chronic, non-healing wounds represent a challenging socio-economic burden, demanding innovative approaches for successful wound management. Resveratrol (RSV) represents a promising therapeutic candidate, but its therapeutic efficacy and clinical applicability have been hampered by its rapid degradation and/or depletion. Herein, RSV was encapsulated into poly(ε-caprolactone) (PCL) microparticles by electrospraying with the aim to prolong and preserve RSV’s release/activity, without affecting its therapeutic properties. Electrospraying led to the fabrication of spherical (2 to 10 μm in size), negatively charged (<−1 mV), and quasi-monodisperse (PDI < 0.3) microparticles, with 60% RSV release after 28 days. Microencapsulation of RSV into PCL prevented its photochemical degradation and preserved its antioxidant properties over 72 h. The RSV-PCL microparticles did not exhibit any cytotoxicity on human dermal fibroblasts. RSV released from the microparticles was biologically functional and induced a significant increase in collagen type I deposition. Furthermore, the produced RSV-PCL microparticles reduced the expression of inflammatory (IL-6, IL-8, COX-2) and proteolytic (MMP-2, MMP-9) mediators. Collectively, our data clearly illustrate the potential of electrosprayed polymeric carriers for the sustained delivery of RSV to treat chronic wounds.

## 1. Introduction

Chronic, non-healing wounds (e.g., diabetic foot ulcers, venous leg ulcers, pressure ulcers) affect millions of patients worldwide and, considering the aging and multimorbid population, represent an increasing challenge and socio-economic burden [1,2,3]. Chronic wounds exhibit an abnormal pattern of wound healing, characterized by a chronic state of inflammation sustained by abundant immune cells infiltration (largely neutrophils and macrophages) [4,5,6]. Dysregulation of pro-inflammatory cytokines (tumor necrosis factor (TNF)-α, interleukin (IL)-1, and IL-6) creates a hostile and highly proteolytic microenvironment with increased matrix metalloproteinase (MMP) activity and thus disproportionate degradation of tissue extracellular matrix (ECM) [7]. Furthermore, excessive production of reactive oxygen species (ROS) results in a pro-oxidant microenvironment that induces severe cell damage and amplifies the expression of proteases and inflammatory mediators [8,9]. Although inflammatory cells are the main source of such molecules, tissue-resident fibroblasts, whose role in ECM assembly and remodeling has been extensively recognized, also secrete an array of growth factors, chemokines, and cytokines (e.g., cyclooxygenase (COX)-2, IL-6, IL-8) which are crucial in the regulation and persistence of the chronic inflammatory response [10,11]. Therefore, modulation of fibroblasts’ behavior could represent an attractive therapeutic strategy for the treatment of chronic wounds [12].

Considering the multifactorial nature of chronic wounds, multifunctional therapeutics capable of targeting multiple aspects of the pathology, including increased oxidative stress, persistent inflammatory stimuli, dysfunctional ECM deposition, and exacerbated MMPs synthesis could represent a successful treatment.

In this view, resveratrol (RSV), a polyphenol and phytoalexin found in large amounts in red wine, has gained increasing interest in the health sector due to its multifactorial health-promoting effects, including its anti-inflammatory, antioxidant, anti-aging, and antimicrobial properties [13,14,15,16]. Furthermore, RSV has been shown to promote cell proliferation, enhance the production of ECM, and reduce the levels of MMPs [17,18,19,20,21], hence possessing a great capacity to interfere with molecular and cellular processes occurring in chronic wounds. Despite the profound potential of RSV, its clinical applications have been hampered by its low bioavailability, sensitivity to light, fast oxidization, and pronounced degradation [22,23,24]. Therefore, strategies to reduce the degradation of RSV and increase its local sustained release and therapeutic efficacy are needed.

Numerous encapsulation techniques for polyphenols have been investigated (e.g., spray-drying [25], freeze-drying [26], emulsification [27]). Nevertheless, these techniques have several drawbacks, mainly due to the harsh fabrication conditions (pH, temperature) that can compromise RSV’s properties and activity. Electrospraying represents a simple, versatile, and gentle electrohydrodynamic encapsulation technique that could overcome these limitations, as it can generate solid carriers in a one-step process without the need for toxic solvents or high temperatures [28,29,30]. Electrospraying is the process of the atomization of a solution into a fine spray of charged droplets upon the application of a high voltage [31]. Progress in electrospraying protocols has made it possible to fabricate polymeric particles with controlled size distributions at much faster production rates [32,33]. Amongst a wide range of polymers that can be used for drug encapsulation, poly(ε-caprolactone) (PCL) has gained increasing interest due to its well-characterized biocompatibility and biodegradability properties and its Food and Drug Administration (FDA) approval for a wide range of clinical uses.

Therefore, the overall goal of this in vitro study was to explore electrospray-based controlled delivery of RSV as a potential therapeutic method for the treatment of chronic wounds, specifically targeting fibroblasts’ behavior. We hypothesized that the encapsulation of RSV by electrospraying into polymeric carriers will result in cytocompatible microparticles with preserved and prolonged RSV activity/release. We further hypothesized that RSV-loaded microparticles will improve ECM deposition of dermal fibroblasts and reduce the expression of pro-inflammatory and proteolytic markers.

## 2. Materials and Methods

### 2.1. Materials

All chemicals and reagents were purchased from Sigma Aldrich (Saint Louis, MO, USA) unless otherwise stated. Tissue culture consumables were purchased from Greiner Bio-One (Monroe, NC, USA).

### 2.2. Fabrication of Electrosprayed Particles

Electrospraying was carried out on a commercial climate-controlled setup (Spraybase^®^, Kildare, Ireland). An amount of 50 mg/mL PCL (average molecular weight (MW) = 72 kDa, Corbion, 1850001, Amsterdam, The Netherlands) was dissolved in chloroform. RSV (Spectrum Chemical, R3116, New Brunswick, NJ, USA) was dissolved in pure ethanol and added at 2.5, 5, and 10 weight % of the total PCL weight. PCL and RSV-PCL solutions were extruded at 0.5 mL/hour through a 26 G stainless steel blunt needle. Upon application of high voltage (16 kV), the solvent evaporated and the electrosprayed particles were collected on a stainless steel dish placed at a 25 cm distance from the needle and filled with 1% polyvinyl alcohol (PVA), dissolved in ultrapure water. The PVA solution was continuously stirred by a magnetic mixer to ensure a homogenous coating and to prevent particle aggregation. All electrospray experiments were carried out at 25 °C and 50% relative humidity. After completion, the particle suspension was centrifuged at 16,000× *g* for 20 min, washed with deionized (DI) water, and centrifuged once more to remove excess surfactant. Lastly, the particles were resuspended in DI water and frozen at −80 °C prior to lyophilization. The dried particles were stored at −20 °C until further use.

### 2.3. Morphological Analysis

The electrosprayed particles were mounted onto carbon disks, gold sputter-coated, and imaged with a VEGA3 TESCAN scanning electron microscope (TESCAN ORSAY HOLDING, Brno, Czech Republic). Particle diameter analysis was conducted using the ImageJ software (Version 1.53q, NIH, Bethesda, MD, USA).

### 2.4. Dynamic Light Scattering (DLS) Analysis

Zeta potential, polydispersity index (PDI), and the hydrodynamic radius of PCL and RSV-loaded PCL particles were assessed using dynamic light scattering (Zetasizer ZS90, Malvern Instruments, Malvern, UK). The samples were prepared in phosphate-buffered saline (PBS) at a concentration of 2 mg/mL and sonicated for 5 min at 40 kHz and 70 W of power in an ultrasonic bath (Fisher Scientific, Waltham, MA, USA) before the analysis to ensure a dispersed solution. 

### 2.5. Fourier-Transform Infrared Spectroscopy (FT-IR) Spectroscopy Analysis

To study the presence of RSV in the polymeric PCL particles, samples were analyzed by FT-IR (IRPrestige-21 FTIR-8400S, Shimadzu, Kyoto, Japan) operating in the attenuated total reflectance (ATR-FTIR) mode. A total of 32 scans were run for each sample at a resolution of 4 cm^−1^ and the spectra were recorded from 400 to 4000 cm^−1^.

### 2.6. Differential Scanning Calorimetry (DSC) and Thermogravimetric (TGA) Analyses

Thermal analyses of the PCL particles and RSV before and after loading were performed using a TA DSC 2500 calorimeter (TA Instruments, New Castle, DE, USA). Samples of 10–15 mg were loaded into hermetic aluminum pans and heated from 25 to 300 °C at a heating rate of 10 °C/min. The endothermic transition of PCL and RSV was recorded as a peak.

The thermogravimetric curves were obtained using a thermobalance TGA 5500 (TA Instruments, New Castle, DE, USA) in the temperature range of 25 to 600 °C. The samples of 10–15 mg were placed onto platinum crucibles and submitted to a heat ramp (2 °C/min) under a dynamic nitrogen atmosphere at a flow rate of 50 mL/min. Calibration was performed beforehand using copper sulphate pentahydrate. Data analysis was performed using the TA Universal Analysis 2000 software (Version 4.7, TA Instruments, New Castle, DE, USA).

### 2.7. Nuclear Magnetic Resonance (NMR) Spectroscopy Analysis

Proton nuclear magnetic resonance (^1^H-NMR) was performed on all the samples to confirm the presence of RSV in the microparticles. Samples of 5 mg were dissolved in 0.5 mL of deuterated dimethyl sulfoxide (DMSO). The NMR spectra were collected on a Bruker Advanced III 500 spectrometer (Bruker, Billerica, MA, USA) with a 20-ppm spectral width centered at about 6 ppm and a total collection size of 65.5 k points. Data analysis of the NMR spectra was performed with Topspin v9.0, and data processing comprised of Fourier transforms, phase/baseline corrections, and chemical shift referencing of the spectra.

### 2.8. Resveratrol Loading Capacity and Encapsulation Efficiency Analyses

RSV-loaded PCL particles were weighed, dissolved in DMSO at a final concentration of 1 mg/mL, and analyzed for absorbance at 330 nm, which is the maximum absorbance wavelength of RSV as determined by a wavelength scan (Appendix A), using a microplate reader (SpectraMax iD3, Molecular Devices, San Jose, CA, USA). PCL in the solution did not affect RSV’s absorbance spectrum (Appendix A; PCL spectrum: Appendix A). Absorbance values were then compared to an 8-point standard curve allowing for the interpolation of the unknown RSV concentration in the sample. The loading capacity was calculated using the equation below:$$\frac{\mathrm{Weight}\;\mathrm{of}\;\mathrm{RSV}\;\mathrm{in}\;\mathrm{microparticles}\;(\mu g)}{\mathrm{Total}\;\mathrm{weight}\;\mathrm{of}\;\mathrm{microparticles}\;(\mathrm{mg})}=\mathrm{Loading}\;\mathrm{capacity}$$

RSV encapsulation efficiency was calculated using the equation below:$$\frac{\mathrm{Amount}\;\mathrm{of}\;\mathrm{RSV}\;\mathrm{in}\;\mathrm{microparticles}\;(\mu g/\mathrm{mg})}{\mathrm{Initial}\;\mathrm{amount}\;\mathrm{of}\;\mathrm{RSV}\;\mathrm{used}\;(\mathrm{mg})}\;\times \;100=\%\;\mathrm{Encapsulation}\;\mathrm{efficiency}$$

### 2.9. Photodegradation Analysis

For the evaluation of the effect of PCL microencapsulation on the photochemical behavior of RSV, 5 mg of microparticles were irradiated with a UV lamp operating at 250 nm at a distance of 10 cm (average intensity: 100 µW/cm^2^). As a control, pure RSV in a powder form was used. At time intervals of 5, 10, 20, and 30 min, specimens were dissolved in DMSO at a final concentration of 1 mg/mL, and RSV was quantified by measuring the absorbance at 330 nm. Resveratrol photodegradation was calculated using the equation below:$$\frac{\left(A0\;-\;A\right)}{A0}\;\times \;100=\%\;\mathrm{Photodegradation}$$
where A_0_ corresponds to the initial absorbance before UV irradiation, and A corresponds to the sample absorbance after UV irradiation.

### 2.10. Antioxidant Capacity Analysis

The antioxidant capacity of the RSV-loaded PCL particles was evaluated by measuring their ability to scavenge the free 2,2-diphenyl-1-picrylhydrazyl (DPPH) radical, using a commercially available kit (Dojindo Molecular Technologies, D678, Rockville, MD, USA). As a control, pure RSV in a powder form was used. Briefly, samples were dissolved in DMSO at a concentration of 1 mg/mL. An amount of 20 µL of the sample was incubated with 500 µL of DPPH working solution. At time intervals of 0.5, 2, 24, 48, and 72 h of incubation at room temperature in the dark, absorbance was measured at 517 nm using a microplate reader (SpectraMax iD3, Molecular Devices, San Jose, CA, USA). Radical scavenging activity (% inhibition) was calculated using the equation below:$$\frac{\left(\mathrm{Anc}\;-\;\mathrm{As}\right)}{\mathrm{Anc}}\;\times \;100=\%\;\mathrm{Radical}\;\mathrm{scavenging}\;\mathrm{activity}\;(\mathrm{inhibition})$$
where A_nc_ corresponds to the absorbance of negative controls and A_s_ corresponds to the absorbance of the samples.

### 2.11. In Vitro Release Profile Analysis

For the release study, 5 mg of particles were resuspended in 0.5 mL of PBS and stored at 37 °C on a shaker set at 150 rpm. At predetermined time points of 1, 6, 12, 24, 72 h, and 7, 14, 21, and 28 days, the samples were centrifuged at 16,000× *g* for 20 min and 300 μL of PBS supernatant was removed and replaced with an equal amount of fresh PBS.

The collected samples were stored at −20 °C and later analyzed using the aforementioned multi-plate reader and settings. Using the acquired absorbance values and comparing them to the initial quantity of RSV in the particles, the cumulative release percentage across all time points was calculated as follows:$$\frac{R1}{R2}\;\times \;100=\%\;\mathrm{Cumulative}\;\mathrm{release}$$
where R_1_ represents the total amount of RSV released in the medium at a given time point, and R_2_ represents the total measured amount of RSV encapsulated in the PCL particles.

### 2.12. Cell Culture

Human adult dermal fibroblasts (hDFs, PCS-201-012, ATCC, Manassas, VA, USA) were routinely sub-cultured and used between passages 3 and 5, with DMEM-F12 (HyClone, SH30023.01, Logan, UT, USA) supplemented with 10% FBS (HyClone, SH30396.02, Logan, UT, USA) and 1% antibiotic/antimycotic (Gibco, 15240-062, Waltham, MA, USA). The medium was changed every 2–3 days. For the various experiments, cells were seeded at 25,000 cells/cm^2^ in 24-well plates and allowed to attach for 24 h, after which the culture media were changed to media containing 100 µg/mL of 0% RSV-PCL or 10% RSV-PCL particles. Media containing 5 μM of RSV were used as a positive control. Samples were analyzed on days 1, 3, and 7. To simulate an inflammatory environment and assess the effect of 0% RSV-PCL and 10% RSV-PCL particles on gene expression, cytokines release, and wound healing, tumor necrosis factor (TNF)-α (PeproTech, 300-01A, Cranbury, NJ, USA) at a final concentration of 50 ng/mL was added to the fibroblast (untreated cells were used as control, CTRL). For the assessment of collagen type I deposition, cells were cultured under macromolecular crowding (MMC) conditions using a cocktail of 37.5 mg/mL Ficoll 70 kDa and 25 mg/mL Ficoll 400 kDa and in combination with or without TNF-α. MMC has been extensively used as a biophysical tool to accelerate extracellular matrix (ECM) deposition in vitro [34,35,36,37]. The addition of macromolecules into the culture media imitates the naturally crowded in vivo milieu, and thus increases the deposition of cell-secreted ECM [38,39].

### 2.13. Cell Viability Analysis

At various time points (1, 3, and 7 days), calcein-AM (Thermo Fisher Scientific, C1430, Waltham, MA, USA) and ethidium homodimer I (Thermo Fisher Scientific, E1169, Waltham, MA, USA) staining was performed, as per the manufacturer’s protocol, to assess the influence of the electrosprayed particles on cell viability. Briefly, cells were washed with PBS, and a solution of calcein-AM (4 μM) and ethidium homodimer I (2 μM) was added. Cells were incubated at 37 °C and 5% CO_2_ for 30 min after which fluorescence images were captured with an Olympus IX-81 inverted microscope (Olympus Corporation, Tokyo, Japan) using the appropriate filter set.

### 2.14. Cell Proliferation Analysis

To assess dermal fibroblasts proliferation, the total viable cell number was calculated by counting the Hoechst 33,342 (Thermo Fisher Scientific, 62249, Waltham, MA, USA) stained nuclei from nine randomly selected images from each group at each given time point (1, 3, and 7 days).

### 2.15. Cell Metabolic Activity Analysis

At various time points (1, 3, and 7 days), the alamarBlue^®^ assay (Thermo Fisher Scientific, DAL1025, Waltham, MA, USA) was used to evaluate the influence of electrosprayed particles on the cellular metabolic activity, as per the manufacturer’s instructions. Briefly, at each time point, cells were washed with PBS and a 10% alamarBlue^®^ solution in PBS was added to the cells. Cells were incubated at 37 °C and 5% CO_2_ for 4 h and the absorbance was measured at 550 nm and 595 nm on the SpectraMax iD3 microplate reader (Molecular Devices, San Jose, CA, USA). Cellular metabolic activity was expressed as a percentage reduction of the alamarBlue^®^ dye and was normalized to untreated controls at each time point.

### 2.16. Sodium Dodecyl Sulphate Polyacrylamide Gel Electrophoresis (SDS-PAGE) Analysis

At various time points (3 and 7 days), the culture medium was aspirated, and cells were briefly washed with PBS. Cells were then digested with pepsin from porcine gastric mucosa at 0.1 mg/mL in 0.5 M acetic acid (Thermo Fisher Scientific, 984303, Waltham, MA, USA) at 37 °C for 2 h under agitation. Cells were then scraped and neutralized with 1 N sodium hydroxide and analyzed by SDS-PAGE under non-reducing conditions using a Mini-Protean 3 system (Bio-Rad Laboratories, Hercules, CA, USA). Bovine collagen type I (100 μg/mL, Advanced Biomatrix, 5005, Carlsbad, CA, USA) was used as the standard for all gels. Staining of the protein bands was performed with the SilverQuest™ kit (Thermo Fisher Scientific, LC6070, Waltham, MA, USA) per the manufacturer’s instructions. To quantify the cell-produced collagen type I deposition, the relative densities of collagen α1 (I) and α2 (I) chains were evaluated with ImageJ and compared to the α1 (I) and α2 (I) chain bands densities of standard collagen type I. Results were normalized to the untreated control-MMC/-TNF-α group at each time point.

### 2.17. Immunocytochemistry Analysis

At various time points (1 or 7 days), cells were briefly washed with PBS and fixed with 4% PFA for 30 min at room temperature. Cells were washed again, and non-specific site interactions were blocked with 3% bovine serum albumin (BSA) in PBS for 1 h. Cells were incubated overnight at 4 °C with the various primary antibodies (rabbit anti-collagen type I (Bio-Techne, NB600-408, Minneapolis, MN, USA), rabbit anti-NF-κB p65 (Cell Signalling Technology, 8242S, Danvers, MA, USA)), after which, they were washed 3 times with PBS, followed by 1 h incubation at room temperature with the secondary antibody (AlexaFluor^®^ 488 goat anti-rabbit, A-11008, Thermo Fisher Scientific, Waltham, MA, USA). F-actin and nuclei were counterstained with Rhodamine phalloidin (Thermo Fisher Scientific, R415, Waltham, MA, USA) and Hoechst 33342 (62249, Thermo Fisher Scientific, Waltham, MA, USA), respectively. Fluorescent images were captured with an Olympus IX-81 inverted microscope. The NF-κB p65 nuclear to cytosolic ratio was calculated, creating a pipeline sequence in the CellProfiler software (Version 4.2.1 https://cellprofiler.org/). The sequence was developed to identify and create a mask of the nuclear area, analyzing the nuclear signal, and a mask of the cytosolic area obtained by subtracting the nuclear area from the total cell area. The nuclear to cytoplasmic ratio (N/C) was then calculated by the software on the NF-κB p65 channel as the ratio between the mean fluorescence intensity of the nuclear mask and the mean fluorescence intensity of the cytosolic area.

### 2.18. RNA Isolation and Gene Expression Analysis (RT-qPCR)

At various time points (1, 3, and 7 days), total RNA was isolated from TRI-Reagent^®^ (Thermo Fisher Scientific, AM9738, Waltham, MA, USA) for 5 min at room temperature to lyse the cells. Afterward, chloroform was added and incubated at room temperature for 5 min while periodically mixing/shaking the sample. The solution was centrifuged at 16,000× *g* for 15 min at 4 °C and the upper aqueous phase containing the RNA was collected. Total RNA was precipitated for 30 min at −20 °C with an equal volume of ice-cold isopropanol, followed by centrifugation for 30 min at 16,000× *g* at 4 °C. RNA pellets were washed in 75% ethanol, air-dried, and resuspended in RNAse-free water. RNA yield and purity were determined using a NanoPhotometer^®^ N60/N50 (Implen, Munich, Germany). An OD 260/280 value between 1.8 and 2.0 was defined as pure RNA. The TaqMan Reverse Transcription kit (Applied Biosystems, 4374966, Waltham, MA, USA) was used to reverse-transcribe 1 µg of RNA into cDNA in a 20 µL volume reaction. An amount of 10 ng cDNA per reaction was subsequently analyzed by quantitative PCR using the TaqMan™ Fast Universal PCR Master Mix (Applied Biosystems, 44-449-64, Waltham, MA, USA) and Predesigned TaqMan^®^ Gene Expression Assays (Applied Biosystems, Waltham, MA, USA). Primer details and assay IDs are listed in Appendix A). Default amplification conditions were as follows: 95 °C for 10 min; 40 cycles of 95 °C for 1 s and 60 °C for 20 s using a QuantStudio^™^ 3 Real-Time PCR Systems (Thermo Fisher Scientific, Waltham, MA, USA). Technical duplicates were measured for each sample and the relative expression level was calculated by the ΔΔCt method using tyrosine 3-monooxygenase/tryptophan 5-monooxygenase activation protein zeta (YWHAZ) as the reference gene. The results are shown as fold change relative to untreated cells at each time point.

### 2.19. Multiplex Enzyme-Linked Immunosorbent Assay (ELISA) Analysis

Conditioned media from hDFs exposed to the different conditions were collected after 7 days of treatment and stored at −80 °C. Cytokine profiles were measured by Quantibody Human Inflammatory Array 1 (RayBiotech, QAH-INF-1-1, Peachtree Corners, GA, USA) which permitted the detection of 10 inflammation-associated cytokines, including interleukin (IL)-1α, IL-1β, IL-4, IL-6, IL-8, IL-10, IL-13, monocyte chemoattractant protein-1 (MCP-1), interferon (IFN)-γ and TNF-α in a single procedure. One standard glass slide was divided into 16 wells of identical cytokine antibody arrays. Each antibody, together with the positive controls, was arrayed in quadruplicate. The protein array slides spotted by specific capture antibodies were incubated with thawed media, washed, and incubated with a cocktail of biotinylated antibodies using the protocol provided by the manufacturer. The slides with bound biotin were then incubated with Cy3 equivalent dye-conjugated streptavidin. Relative fluorescent strength was detected by the InnoScan 710 microarray scanner (Innopsys, Chicago, IL, USA). The results are shown as fold change release relative to untreated cells.

### 2.20. Statistical Analysis

Data are expressed as mean ± standard deviation. All experiments were conducted at least in triplicates. When assumptions were fulfilled (normality, variance homogeneity), ANOVA with Tukey’s post hoc test was used for pairwise comparisons. Distribution was tested for normality using the Anderson–Darling test, and Bartlett’s and Levene’s tests were used for homogeneity of variances. In cases that followed gamma distributions, generalized linear model-based gamma distributions with log-link function were used to test hypotheses globally. Due to the small sample sizes, corresponding multiple comparisons were done using one sample bootstrap-*t*-tests—for comparisons against the reference values—and two-sample bootstrap-*t*-tests were used. When data could not be analyzed semi-parametrically, non-parametric analysis was conducted using the Kruskal–Wallis test for multiple comparisons and the Mann–the Whitney test for pairwise comparisons. All hypotheses were tested two-sided with a significance level of 5%. The statistical significance was accepted at *p* < 0.05. The analyses were done with MATHEMATICA 12 (Wolfram Research, Inc., Mathematica, Version 12, Champaign, IL, USA (2019)) and MINITAB^®^ version 19 (Minitab Inc., State College, PA, USA).

## 3. Results

### 3.1. Morphological and Physicochemical Analyses of Electrosprayed Microparticles

Scanning electron microscopy analysis (0% RSV-PCL Figure 1A, 2.5% RSV-PCL Figure 1B, 5% RSV-PCL Figure 1C, 10% RSV-PCL Figure 1D) revealed that all electrosprayed samples were composed of uniform (fiber-free) particles spherical in shape. Electrosprayed particles produced without PVA coating exhibited a similar morphology (Appendix A). Particle size distribution analysis showed that the 0% RSV-PCL (Figure 1E), 2.5% RSV-PCL (Figure 1F), and 5% RSV-PCL (Figure 1G) samples were comprised of particles with a diameter range from 5 µm to 10 µm, while the 10% RSV-PCL (Figure 1H) samples were comprised of particles with a diameter range from 2 µm to 4 µm and were significantly (*p* < 0.01) smaller in comparison to the other groups. However, no statistically significant (*p* > 0.05) differences were observed in hydrodynamic radii among all microparticles (Figure 1I). All RSV-loaded microparticles exhibited lower PDI in comparison to 0% RSV-PCL particles (Figure 1J), albeit not statistically significant (*p* > 0.05). Zeta potential analysis revealed that all electrosprayed microparticles were of a negative charge and no statistically significant (*p* > 0.05) differences were observed between groups (Figure 1K).

The control PCL, 0%, 2.5%, 5%, 10% RSV-PCL particles, and control RSV powder were analyzed by FT-IR to identify the chemical composition of the microparticles and to reveal the presence of RSV in the formulation mixtures (Figure 2A). FT-IR spectra for 2.5%, 5%, and 10% RSV-PCL particles revealed the presence of characteristic bands at 3250 cm^−1^ corresponding to O–H stretching of RSV, thus confirming its presence in the PCL particles. A slight shift in maximum transmittance is observed for the O–H stretching band upon RSV encapsulation, indicating hydrogen bonding between RSV and PCL. Differential scanning calorimetry (DSC) analysis (Figure 2B) of control RSV powder revealed a sharp endothermic peak at 260 °C, corresponding to the melting temperature of its crystalline form. DSC of control PCL, 0%, 2.5%, 5%, and 10% RSV-PCL particles, showed an endothermic peak at 50–70 °C, which corresponds to the melting temperature of the crystalline portions of PCL. In the majority of tests, 2.5%, 5%, and 10% of RSV-PCL particles’ DSC thermograms did not show an endothermic peak at 260 °C, indicating that RSV became fully amorphous after the electrospraying process. Only in rare cases, negligible amounts of crystalline RSV could be detected at 260 °C (Appendix A). This may be due to the adsorption of RSV to the particle surface, which can be attributed to the poor solubility and compatibility of RSV in chloroform (used to dissolve PCL). Thermogravimetric (TGA) analysis (Appendix A) displayed a weight loss of RSV of 50–60% from 250 °C to 600 °C. The weight losses of RSV-loaded PCL particles started from around 350 °C due to thermal degradation of PCL and reached 100% weight loss at approximately 450 °C. The shift in RSV weight loss upon incorporation in PCL indicates that the polymer enhanced the thermal stability of RSV.

NMR analysis (2.5% RSV-PCL Figure 2C, 5% RSV-PCL Figure 2D, 10% RSV-PCL Figure 2E) revealed six narrow signals in the low field region, above 6 ppm, typical of RSV spectra. NMR spectrum of control RSV powder is provided in Appendix A. The four characteristic signals between 1 ppm and 4 ppm correspond to PCL, as previously reported [40,41].

### 3.2. Drug Content, Drug Release, Photodegradation and Antioxidant Analyses

Loading capacity analysis (Figure 3A) revealed that 10% of RSV-PCL had the highest (*p* < 0.05) loading capacity. No statistically significant (*p* > 0.05) differences were observed in the encapsulation efficiency among all microparticles (Figure 3B). Cumulative release analysis (Figure 3C) revealed that the 10% RSV-PCL particles presented slower, although not statistically significant (*p* > 0.05), drug release, reaching a 60% cumulative release after 28 days. These findings could be explained by assuming that, in a lower polymer-drug ratio, RSV molecules could be less dispersed in the PCL matrix, diffusing at a slower rate. Furthermore, decreasing the polymer amount, with respect to the drug being loaded, can cause the formation of a more impermeable structure, which impedes water diffusion, thereby decreasing the dissolution rate. Overall, all three groups demonstrated sustained and controlled release of RSV, confirming the protective effect of the PLC microparticles. DPPH radical scavenging activity analysis (Figure 3D) revealed that at 0.5, 2, 24, and 48 h, pure RSV powder had the highest (*p* < 0.05) antioxidant effect, while the 2.5% RSV-PCL microparticles had the lowest (*p* < 0.05) effect among the microparticles tested. At 72 h, no statistically significant (*p* > 0.05) differences in antioxidant capacity were observed between pure RSV powder and 10% RSV-PCL microparticles. Overall, RSV-loaded microparticles showed an increase in antioxidant capacity as a function of time, while pure RSV lost its antioxidant properties over time. Photodegradation analysis (Figure 3E) showed that RSV encapsulation in PCL microparticles significantly (*p* < 0.05) reduced photodegradation at all time points in comparison to pure RSV powder control.

### 3.3. Cell Attachment, Viability, Metabolic Activity, and Proliferation Analyses

Cell attachment analysis (Figure 4A) revealed that 0% RSV-PCL and 10% RSV-PCL microparticles did not negatively affect human dermal fibroblasts attachment as a function of time. Viability analysis (Figure 4B) through Live/Dead assay showed that 0% RSV-PCL and 10% RSV-PCL did not significantly alter human dermal fibroblasts’ viability as a function of time. Metabolic activity analysis (Figure 4C) through the alamarBlue^®^ assay showed no toxicity of 0% RSV-PCL and 10% RSV-PCL on human dermal fibroblasts as a function of time. Proliferation analysis (Figure 4D), through the counting of Hoechst-stained nuclei, revealed that 0% RSV-PCL and 10% RSV-PCL induced an increase in the number of human dermal fibroblasts as a function of time, similar to untreated control cells.

### 3.4. Collagen Type I Deposition Analysis

SDS-PAGE (Figure 5A) and corresponding densitometric analysis (Figure 5B) demonstrated that MMC induced a significant (*p* < 0.05) increase in collagen type I deposition in all conditions at all time points, independent of TNF-α stimulation. At all time points, RSV-PCL microparticles, in the presence of TNF-α stimulation and MMC conditions, induced the highest (*p* < 0.05) collagen type I deposition in comparison to all other groups. Collagen matured as a function of time in culture, as evidenced by the presence of β-bands. Immunocytochemistry (Figure 5C) of collagen type I confirmed that 10% RSV-PCL microparticles and TNF-α stimulation under MMC conditions induced the highest collagen type I deposition.

### 3.5. Gene Expression, Multiplex ELISA and p65 Localization Analyses

Gene expression (Figure 6) analysis revealed that, at all time points, 10% RSV-PCL microparticles significantly (*p* < 0.05) reduced the expression of the inflammatory mediators IL-6, IL-8, and COX-2 in comparison to cells treated with TNF-α. With regards to ECM remodeling enzymes, 10% RSV-PCL microparticles significantly (*p* < 0.05) reduced the expression of MMP-2 on day 1 and day 3, and MMP-9 on day 1 and day 7, in comparison to cells treated with TNF-α. No statistical differences (*p* > 0.05) were observed in the gene expression of collagen type I. However, on day 3 and day 7, 10% RSV-PCL microparticles significantly (*p* < 0.05) reduced the expression of α-SMA. No statistical differences (*p* > 0.05) were observed in the gene expression of MMP-1, MMP-3, MMP-13, TIMP-1, and ADAMTS4 (Appendix A). A detailed statistical analysis of gene expression results is listed in Appendix A.

Multiplex ELISA (array setup: Appendix A, array scans: Appendix A) and complementary quantification (Figure 7) analysis showed that 10% RSV-PCL microparticles significantly (*p* < 0.05) reduced fibroblasts secretion of IL-6, IL-8, and MCP-1. No statistical differences (*p* > 0.05) were observed in IL-1α, IL-1β, IL-4, IL-10, IL-13, IFN-γ and TNF-α.

Immunocytochemistry (Figure 8A) and complementary relative fluorescence intensity (Figure 8B) analysis of the NF-κB subunit p65 illustrated that the stimulation of fibroblasts with TNF-α induced significant (*p* < 0.05) nuclear translocation of p65. However, treatment of TNF-α stimulated cells, either with RSV or 10% RSV-PCL microparticles, did not robustly prevent or reverse the nuclear translocation of the NF-κB subunit p65, even though a minor effect was noted (RSV: *p* = 0.041; 10% RSV-PCL: *p* = 0.014).

## 4. Discussion

Given the complex etiology of chronic wounds, RSV represents a multifunctional therapeutic capable of tackling numerous aspects of the wound microenvironment, especially the uncontrolled and self-sustaining inflammatory, proteolytic, and oxidative mechanisms, potentially leading to an effective reparative and regenerative treatment. Nevertheless, the success of resveratrol formulations has thus far been limited by their low bioavailability and pronounced degradation, posing the need to explore methods to increase its therapeutic efficacy and clinical applicability. Considering that electrospraying has emerged as a simple and easily controllable technique to encapsulate therapeutic molecules without any physicochemical degradation or alteration in their function, herein, we ventured to assess whether RSV-loaded electrosprayed microparticles can modulate the ECM deposition and inflammatory response of dermal fibroblasts for the treatment of chronic wounds.

Our results clearly demonstrate that reproducible PCL spheres were produced by electrospraying, with narrow quasi-monodisperse size distributions. PCL was chosen as the polymeric carrier due to its biocompatibility, biodegradability, low immunogenicity, and documented sustained drug release in many delivery systems, used in many FDA-approved medical devices for a wide range of therapeutic applications [42,43,44]. The average size of the PCL particles ranged from 2 to 10 μm, which is in agreement with previous studies that utilized electrospraying for the fabrication of PCL microspheres [45]. Compared to unloaded PCL particles, the size of the 10% RSV-loaded microparticles was reduced, potentially due to the strong interaction between RSV and PCL, resulting in a compaction of the particle core. All microparticles exhibited a slightly negative surface charge, as indicated by their zeta potential. This is an important factor, as particles with negative zeta potential generate repulsive forces and have weak interactions with cells, causing no damage to cell membranes, as opposed to positively charged particles that can cause severe cytotoxicity [46].

The highest loading capacity (26 µg of RSV per mg of PCL) could be achieved in the 10% RSV-PCL particles, which were used for subsequent cell behavior assessments. Loading capacity results are comparable to previous studies using porous scaffolds composed of poly(lactide-co-glycolide) (PLGA) particles loaded with RSV [47] and are three-fold higher than the previously reported RSV loading capacity in biodegradable PLGA sintered microsphere scaffolds [48]. Although higher RSV encapsulation efficiencies have been reported in protein- and polysaccharide-based electrosprayed particles, their translational success is likely limited by particularly fast release profiles. While we achieved the distinct sustained release of RSV (60% after 28 days), these zein- and levan-based particles released 58% within 9 h [49] and 100% after only 7 h [50], respectively. This is of relevant interest, considering that wound chronicity has been defined in the range of 4 weeks to 3 months [51,52]. Furthermore, to promote optimal healing, the frequency of wound dressing changes should be limited (e.g., to once or twice per week) [53,54], thus requiring long-lasting therapeutic effects which are not achievable with fast-releasing products. To further increase the RSV encapsulation efficiency while maintaining suitable release profiles, future studies will focus on the optimization of coaxial electrospraying settings, which could offer more precise control of the core-shell architecture, enabling a more efficient encapsulation [55,56,57]. It is also worth noting the efficient protective function of the PCL microparticles which prevented RSV from photochemical degradation, extensively documented in the literature [58,59], and preserved its antioxidant properties over a longer period of time. Given the positive effect of RSV-microencapsulation, future challenges will be to incorporate RSV-PCL microparticles into a suitable matrix, either in the form of hydrogel or fibrous membrane, to create innovative bioactive dressings with sustained drug release into the wound bed, ensuring tissue retention and reducing particle displacement and damage.

RSV released from the microparticles was biologically functional and induced a significant increase in collagen type I deposition and crosslinking from dermal fibroblasts. This is in agreement with previous studies that showed the positive effect of RSV on collagen assembly and organization through the activation of SIRT1 signaling pathways [60,61,62]. Furthermore, results from gene expression experiments provided clear evidence that RSV-PCL microparticles can effectively reduce mRNA levels of major matrix-degrading enzymes (MMP-2, MMP-9), which are key factors in the progression of chronic wounds, as previously reported [20]. Therefore, RSV-PCL microparticles could have the potential to induce a shift from proteolysis towards regeneration, thus restoring physiological wound repair. It is also worth noting that RSV-PCL microparticles did not affect collagen type I gene expression but reduced α-SMA expression over time. This is significant, considering that the overexpression of these two genes could lead to a progression beyond the initial chronic inflammation and development of an aggressive fibrotic state [63,64]. 

While the role of fibroblasts during ECM assembly and remodeling is well assessed, their contribution to maintaining an inflammatory environment in chronic wounds and the therapeutic potential to regulate their inflammatory phenotype has been investigated far less. Many studies demonstrated that fibroblasts actively condition the local cellular and cytokine microenvironment by secreting a plethora of factors that regulate the nature of the inflammatory infiltrate and wound chronicity [10,11]. We demonstrated that RSV-PCL microparticles significantly reduced the gene expression and paracrine signaling of critical pro-inflammatory cytokines. Furthermore, RSV-loaded microparticles reduced the secretion of MCP-1, an important chemoattractant for monocytes infiltration [65,66], thereby potentially impeding the recruitment of pro-inflammatory macrophages and fibroblasts-macrophages’ activation [67]. Importantly, in vivo animal experiments planned in the future will help to investigate the effect of RSV-loaded microparticles on other cell types, such as immune cells as well as their influence on fibroblasts-immune cells crosstalk.

Given the potent anti-inflammatory effect of RSV, we ventured to determine the involvement of the NF-κB signaling pathway by quantifying nuclear translocation of p65. We demonstrated that the stimulation of dermal fibroblasts with TNF-α resulted in the translocation of the cytosolic p65 into the nucleus, indicating NF-κB activation, as previously reported [68,69,70]. However, treatment with RSV-PCL did not block the nuclear translocation of p65. This is in agreement with a previous publication that showed that RSV blocked the ubiquitination of NEMO and inhibited IκB kinaseβ-mediated NF-κB activation rather than inhibiting p65 nuclear accumulation [71]. Considering that Toll-like receptors (TLRs) trigger the activation of NF-κB [72] and non-healing wounds have been associated with the persistent activation of TLR2/4 and inflammatory cytokine expressions [73,74], future studies should assess the effect of RSV-PCL microparticles on these mechanisms, taking into account that RSV has been shown to inhibit TLRs-mediated signaling pathways [75,76]. Further investigations should not only verify these processes but also focus on the assessment of the mitogen-activated protein kinase (MAPK) pathway since RSV has been shown to decrease the expression of p38 kinase and extracellular signal-regulated kinases (ERK) phosphorylation [77,78,79].

## 5. Conclusions

RSV is a promising compound for the treatment of chronic wounds. However, it is prone to rapid degradation, which hinders the clinical translation of RSV-based therapeutics. Electrospraying has emerged as a fast, simple, and versatile particle fabrication technique to encapsulate therapeutic molecules without jeopardizing their physicochemical properties or altering their function. Herein, electrosprayed PCL microparticles loaded with RSV allowed for the sustained release of RSV while preserving its biological activity. Since RSV-PCL microparticles increased ECM deposition and inhibited the expression of pro-inflammatory and matrix-degrading enzymes in human dermal fibroblasts, they may constitute an efficacious wound therapy by interfering with the vicious cycle of inflammation, proteolysis, and inadequate repair and regeneration in chronic wounds (Figure 9). Future studies will focus on comprehensively assessing the biological activity of the RSV-PCL microparticle-based drug delivery system to determine the optimal administration mode and duration of its therapeutic activity.

## Figures and Tables

**Figure 1 pharmaceutics-14-00853-f001:**
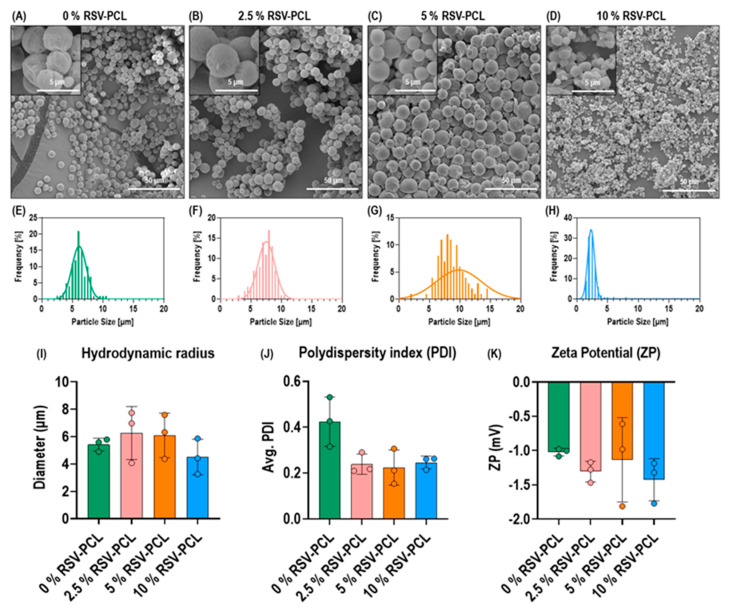
Morphological and dynamic light scattering (DLS) analyses of RSV-PCL electrosprayed microparticles. Scanning electron microscopy analysis of (**A**) 0% RSV-PCL, (**B**) 2.5% RSV-PCL, (**C**) 5% RSV-PCL, (**D**) 10% RSV-PCL revealed that all electrosprayed samples were composed of uniform (fiber-free) particles spherical in shape (*n* = 3 batches). Particles size distribution analysis showed that (**E**) the 0% RSV-PCL, (**F**) 2.5% RSV-PCL, and (**G**) 5% RSV-PCL samples were comprised of particles with a diameter range from 5 µm to 10 µm, while (**H**) the 10% RSV-PCL samples were comprised of particles with a diameter range from 2 µm to 4 µm (*n* = 3 batches). No significant (*p* > 0.05) differences were observed in (**I**) hydrodynamic radii, (**J**) polydispersity indices, and (**K**) zeta potential between groups (*n* = 3 batches). Individual values are displayed as dots on bar charts. Error bars represent mean ± SD. Generalized linear models and bootstrap-*t*-tests.

**Figure 2 pharmaceutics-14-00853-f002:**
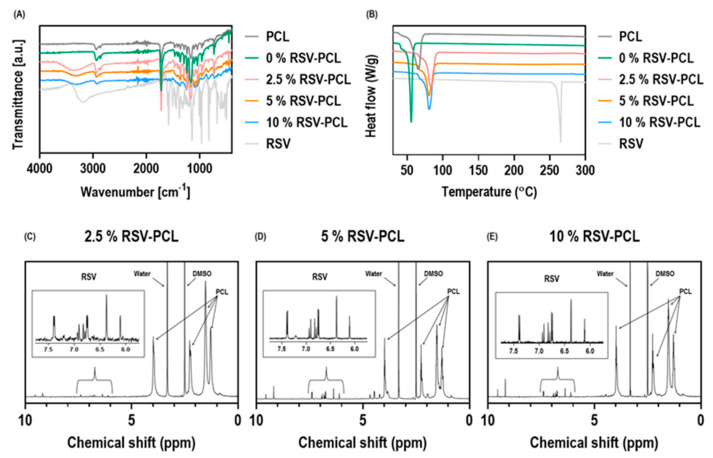
Physicochemical analysis of RSV-PCL electrosprayed microparticles. FT-IR analysis (**A**) revealed the presence of characteristic bands at 3250 cm^−1^ corresponding to O–H stretching of the RSV, thus confirming its presence in the PCL microparticles (*n* = 3 batches). DSC analysis (**B**) of the majority of RSV-loaded microparticle batches showed no endothermic peak at 260 °C which is characteristic of crystalline RSV, indicating that RSV became fully amorphous after electrospraying. ^1^H-NMR analysis of (**C**) 2.5% RSV-PCL, (**D**) 5% RSV-PCL, and (**E**) 10% RSV-PCL revealed 6 narrow signals in the low field region, above 6 ppm, typical of RSV spectra (*n* = 3 batches).

**Figure 3 pharmaceutics-14-00853-f003:**
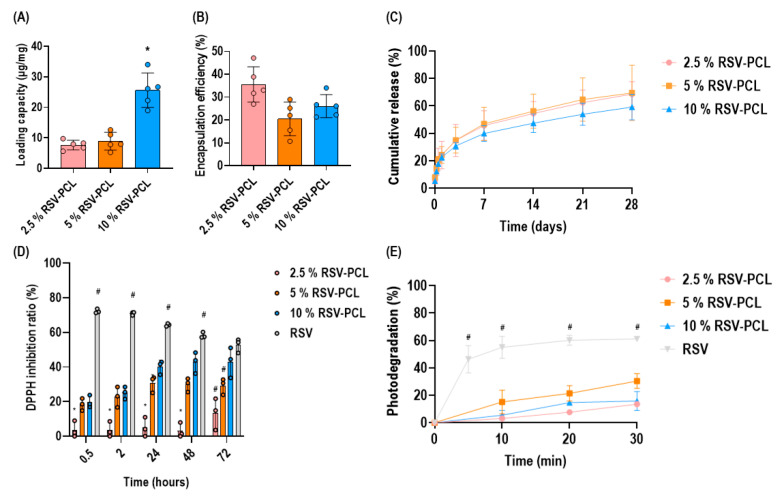
Drug content, drug release, photodegradation, and antioxidant analyses of RSV-PCL electrosprayed microparticles. Loading capacity analysis (**A**) revealed that 10% RSV-PCL had the highest (*p* < 0.05) loading capacity (*n* = 3 batches). Encapsulation efficiency analysis (**B**) showed no statistically significant (*p* > 0.05) differences among all microparticles. Cumulative release analysis (**C**) revealed that all microparticles reached approximately 60% cumulative release after 28 days (*n* = 3 batches). DPPH radical scavenging activity analysis (**D**) revealed that RSV-loaded microparticles showed an increase in antioxidant capacity as a function of time while pure RSV lost its antioxidant properties over time (*n* = 3 batches). Photodegradation analysis (**E**) showed that PCL microparticles significantly (*p* < 0.05) reduced RSV photodegradation at all time points in comparison to pure RSV powder control (*n* = 3 batches). * indicates a statistically significant difference among RSV-PCL microparticles (*p* < 0.05). # indicates a statistically significant difference between pure RSV powder and RSV-PCL microparticles (*p* < 0.05). Individual values are displayed as dots on bar charts. Error bars represent mean ± SD. Generalized linear models and bootstrap-*t*-tests.

**Figure 4 pharmaceutics-14-00853-f004:**
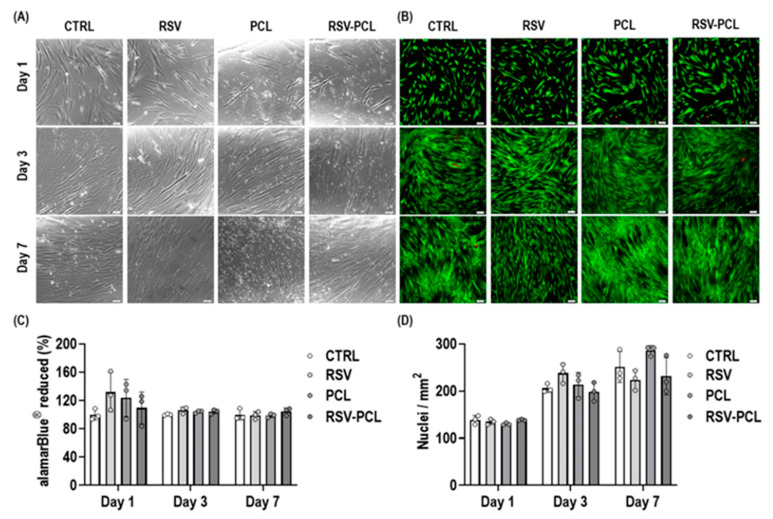
Biocompatibility analysis of 0% RSV-PCL (PCL) and 10% RSV-PCL (RSV-PCL) electrosprayed microparticles on human dermal fibroblasts. Cell attachment (**A**), Live/Dead viability assay (live: green, dead: red. Scale bars: 100 μm) (**B**), alamarBlue^®^ measure of metabolic activity (results normalized to untreated control, CTRL, at each time point, Scale bars: 100 μm) (**C**), and proliferation (**D**) were not affected by PCL and RSV-PCL (100 µg/mL) microparticles as a function of time (1, 3, and 7 days) (*n* = 3). Media containing 5 μM RSV was used as a positive control. Individual values are displayed as dots on bar charts. Error bars represent mean ± SD. Generalized linear models and bootstrap-*t*-tests.

**Figure 5 pharmaceutics-14-00853-f005:**
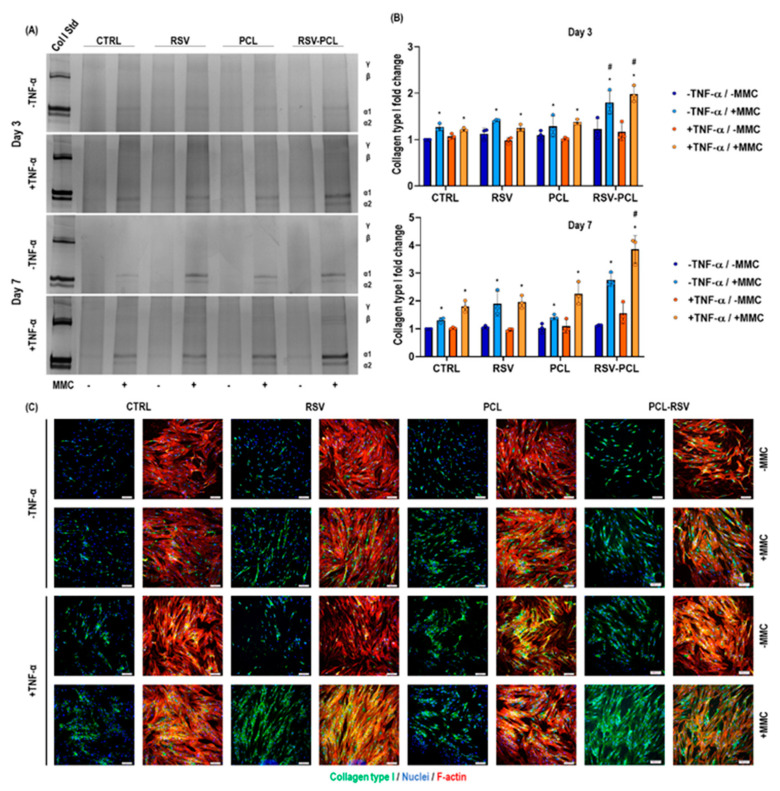
Effect of 0% RSV-PCL (PCL) and 10% RSV-PCL (RSV-PCL) electrosprayed microparticles on collagen type I deposition of human dermal fibroblasts. SDS-PAGE (**A**) and complementary densitometric analysis (**B**) revealed that at all time points (3 and 7 days), RSV-PCL microparticles (100 µg/mL), in the presence of TNF-α stimulation (50 ng/mL) and MMC conditions (Ficoll 70 kDa: 37.5 mg/mL; Ficoll 400 kDa: 25 mg/mL), induced the highest (*p* < 0.05) collagen type I deposition (*n* = 3). Collagen matured as a function of time in culture, as evidenced by the presence of β-bands. Immunocytochemistry analysis (**C**) of collagen type I confirmed that RSV-PCL microparticles and TNF-α stimulation under MMC conditions induced the highest collagen type I deposition after 7 days (*n* = 3. Collagen type I: green, F-actin: red, Nuclei: blue. Scale bars: 100 μm. Media containing 5 μM RSV was used as a positive control. Results were normalized to the untreated control (CTRL)-MMC/-TNF-α group at each time point. * indicates a statistically significant difference to -MMC/-TNF-α CTRL group at each time point (*p* < 0.05). # indicates a statistically significant difference among all groups at each time point (*p* < 0.05). Individual values are displayed as dots on bar charts. Error bars represent mean ± SD. Generalized linear models and bootstrap-*t*-tests.

**Figure 6 pharmaceutics-14-00853-f006:**
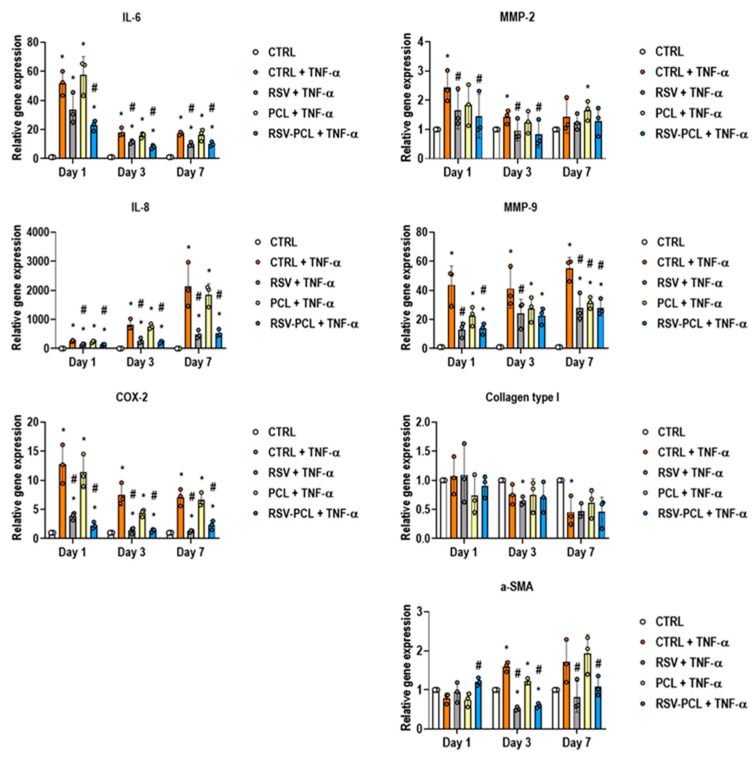
Effect of 0% RSV-PCL (PCL) and 10% RSV-PCL (RSV-PCL) electrosprayed microparticles on gene expression of human dermal fibroblasts. RT-qPCR analysis (*n* = 3) revealed that, at all time points (1, 3, and 7 days), RSV-PCL microparticles (100 µg/mL) significantly (*p* < 0.05) reduced the expression of the inflammatory mediators IL-6, IL-8, COX-2 in comparison to cells treated with TNF-α (50 ng/mL). RSV-PCL microparticles significantly (*p* < 0.05) reduced the expression of the ECM remodeling enzymes MMP-2 (at 1 and 3 days) and MMP-9 (at 1 and 7 days). No statistical differences (*p* > 0.05) were observed in the expression of collagen type I, whereas RSV-PCL microparticles significantly (*p* < 0.05) reduced the expression of α-SMA (at 3 and 7 days). Media containing 5 μM RSV were used as positive controls. The results were normalized to untreated control (CTRL) cells at each time point. * indicates a statistically significant difference to the CTRL group at each time point (*p* < 0.05). # indicates a statistically significant to CTRL + TNF-α group at each time point (*p* < 0.05). Individual values are displayed as dots on bar charts. Error bars represent mean ± SD. Generalized linear models and bootstrap-*t*-tests.

**Figure 7 pharmaceutics-14-00853-f007:**
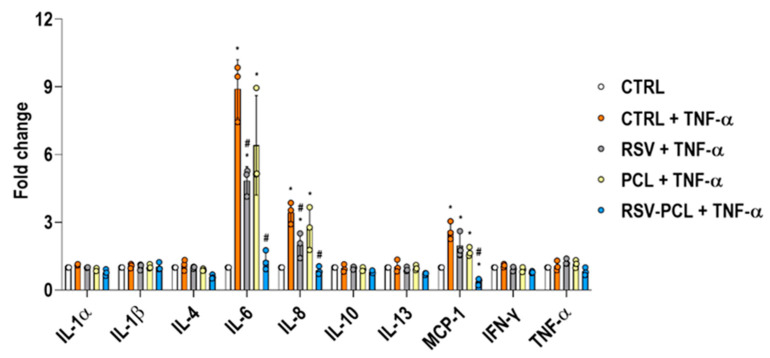
Effect of 0% RSV-PCL (PCL) and 10% RSV-PCL (RSV-PCL) electrosprayed microparticles on inflammatory cytokines release of human dermal fibroblasts. Multiplex ELISA quantification analysis showed that 10% RSV-PCL microparticles (100 µg/mL) significantly (*p* < 0.05) reduced secretion of IL-6, IL-8, and MCP-1 after 7 days of treatment. Media containing 5 μM RSV were used as positive controls. The results are normalized to untreated control (CTRL) cells. * indicates a statistically significant difference to the CTRL group (*p* < 0.05). # indicates statistically significant to CTRL + TNF-α group (*p* < 0.05). Individual values are displayed as dots on bar charts. Error bars represent mean ± SD. Generalized linear models and bootstrap-*t*-tests.

**Figure 8 pharmaceutics-14-00853-f008:**
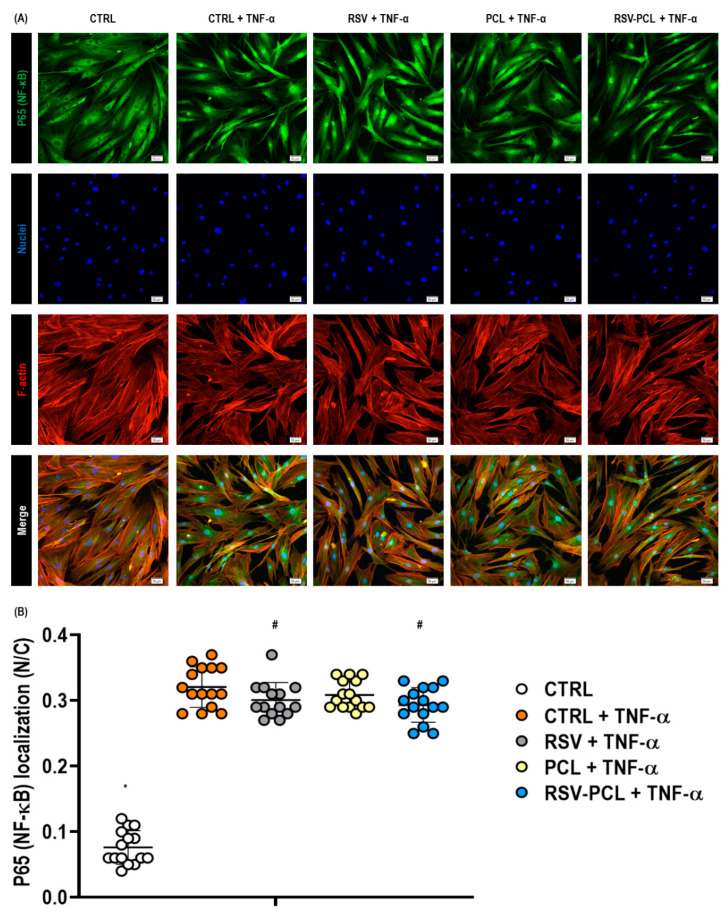
Effect of 0% RSV-PCL (PCL) and 10% RSV-PCL (RSV-PCL) electrosprayed microparticles on the NF-κB signaling pathway. Immunocytochemistry (**A**) and complementary relative fluorescence intensity (**B**) analysis of the NF-κB subunit p65 localization illustrated that TNF-α stimulation (50 ng/mL) induced significant (*p* < 0.05) nuclear translocation of p65 (indicative of NF-κB activation) after 24 h (*n* = 3. p65: green, F-actin: red, Nuclei: blue. Scale bars: 50 μm). The 10% RSV-PCL microparticles did not robustly prevent or reverse nuclear translocation of the NF-κB subunit p65, even though a minor effect was noted (*p* < 0.05). * indicates a statistically significant difference between all groups (*p* < 0.05). # indicates statistically significant to CTRL + TNF-α group (*p* < 0.05). Individual values (*n* = 15 images per group) are displayed as dots on bar charts. Error bars represent mean ± SD. Generalized linear models and bootstrap-*t*-tests.

**Figure 9 pharmaceutics-14-00853-f009:**
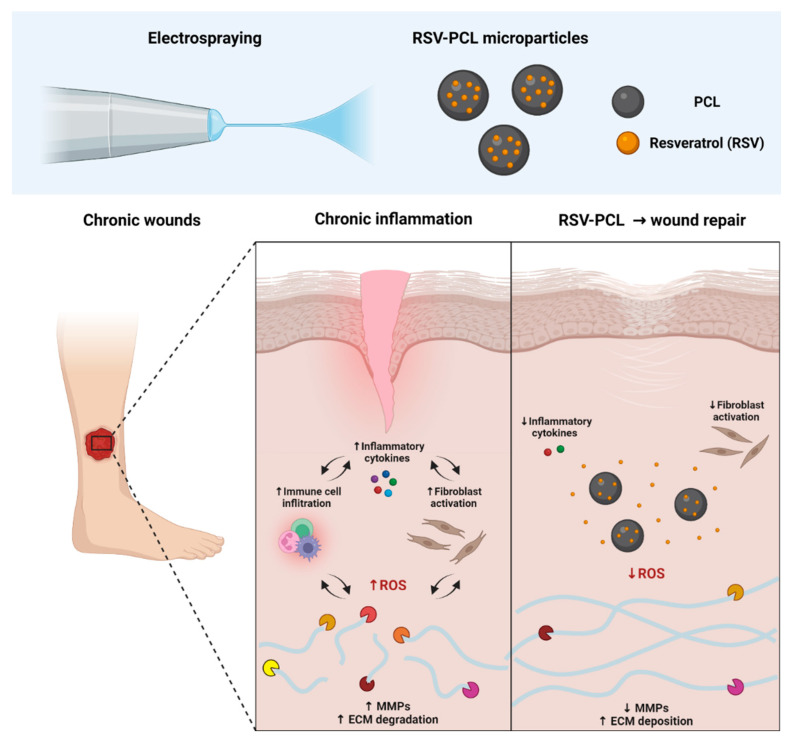
Chronic wounds are characterized by a chronic state of inflammation sustained by abundant immune cell infiltration, which increases the presence of inflammatory cytokines and creates a highly pro-oxidant microenvironment. This leads to the activation of tissue-resident fibroblasts, which contribute to the persistence of local inflammation by further secreting pro-inflammatory factors. Furthermore, excessive production of matrix metalloproteinases (MMPs) generates a highly proteolytic microenvironment, degrading structural components of the extracellular matrix (ECM). Controlled and sustained delivery of RSV through electrosprayed polymeric carriers may represent a valuable therapeutic method for the treatment of chronic wounds, specifically targeting the vicious cycle of inflammation, proteolysis, and inadequate repair. Created with BioRender.com (accessed on 4 April 2022).

## Data Availability

The data presented in this study are available upon request from the corresponding author.

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
