# Peer review of "Resveratrol Microencapsulation into Electrosprayed Polymeric Carriers for the Treatment of Chronic, Non-Healing Wounds"

_pharmaceutics, 2022, doi:10.3390/pharmaceutics14040853_

Round 1

Reviewer 1 Report

The authors aim to explore electrospray-based controlled delivery of RSV as a potential therapeutic method for chronic wound treatment. The RSV was encapsulated into poly(ε-caprolactone) (PCL) to form microparticles by electrospraying. The authors evaluated the release profile of RSV from PCL microparticles and the stability of RSV in RSV-PCL microparticles. And they investigated the treatment effects of RSV-PCL microparticles on human dermal fibroblasts. The results indicated that the encapsulation of RSV into PCL could prolong and preserve RSV’s release/activity. The in vitro results also demonstrated a favorable anti-inflammatory effect of RSV-PCL microparticles in cell experiments. Besides, the RSV released from PCL microparticles increased collagen type I deposition and reduced the expression of proteolytic mediators. These new findings will inspire researchers to further investigate the in vivo therapeutic potential of the RSV-loaded microparticles.

The experiments were well-designed, and the data were well presented. There is one concern about the discussion.

Lines 567-569, the authors claimed that “Compared to unloaded PCL particles, the size of the RSV-loaded microparticles was reduced…”. However, as shown in figure 1, there is an increase in the size of 2.5 % RSV-PCL and 5 % RSV-PCL. The authors should explain this phenomenon.

Author Response

Reviewer No 1:

Comment:Lines 567-569, the authors claimed that “Compared to unloaded PCL particles, the size of the RSV-loaded microparticles was reduced…”. However, as shown in figure 1, there is an increase in the size of 2.5 % RSV-PCL and 5 % RSV-PCL. The authors should explain this phenomenon.

Response: The sentence has been rephrased: Compared to unloaded PCL particles, the size of the 10 % RSV-loaded microparticles was reduced, potentially due to the strong interaction between RSV and PCL, resulting in a compaction of the particle core.

Reviewer 2 Report

This study describes the development of electrosprayed polymeric microparticles based PCL, and their chronic wound healing properties after encapsulation with resveratrol. The resveratrol-loaded PCL microparticles exhibited sustained release and maintained its anti-oxidant properties. The human dermal fibroblast incubated with resveratrol-loaded microparticles induced collagen type 1 deposition and reduced the expression of expression of inflammatory cytokines. Overall, the experiments and results for this study is well organized and the manuscript can be accepted for publication. The reviewer has minor comments for the authors.

  1. The authors abbreviated MMC twice.
  2. The need to re-check the loading efficiency of resveratrol to PCL microparticles. In general, increasing concentration of drugs to microparticles shows decrease in loading efficiency. This was not observed in this case and should be clarified.

Author Response

Reviewer No 2:

Comment:The authors abbreviated MMC twice

Response: The sentence has been rephrased: MMC has been extensively used as a biophysical tool to accelerate extracellular matrix (ECM) deposition in vitro

Comment:The need to re-check the loading efficiency of resveratrol to PCL microparticles. In general, increasing concentration of drugs to microparticles shows decrease in loading efficiency. This was not observed in this case and should be clarified.

Response: The loading capacity has been calculated as follows:

Therefore, the loading capacity is the amount of drug loaded per unit weight of the nanoparticle, indicating the percentage of the mass of the nanoparticle that is due to the encapsulated drug.

We demonstrated that increasing the concentration of resveratrol led to higher loading capacity, which is in agreement with previous publications ( https://www.ncbi.nlm.nih.gov/pmc/articles/PMC7598274/).

On the other hand, the encapsulation efficiency has been calculated as follows:

We actually observed a lower encapsulation efficiency as a function of increasing concentration of resveratrol, which is in agreement with previous publications (https://www.ncbi.nlm.nih.gov/pmc/articles/PMC7598274/). We also discuss potential techniques (e.g., co-axial electrospraying) to increase the encapsulation efficiency of our system.

Reviewer 3 Report

In the manuscript entitled “Resveratrol microencapsulation into electro sprayed polymeric carriers for the treatment of chronic, non-healing wounds”. The authors presented the manuscript in a good format. but there are a few suggestions as follows-

1. The author mentioned that Resveratrol was encapsulated into poly(caprolactone) (PCL) microparticles by electrospraying method, but the author didn’t mention the role of PCL and why they are chosen PCL, the author needs to provide the details in the introduction part.

  1. The author needs to compare the present work with other PCL-based microparticles based on the literature.
  2. Please check for grammar corrections.

Author Response

Reviewer No 3:

Comment:The author mentioned that Resveratrol was encapsulated into poly(caprolactone) (PCL) microparticles by electrospraying method, but the author didn’t mention the role of PCL and why they are chosen PCL, the author needs to provide the details in the introduction part”.

Response: We included the role of PCL and the reason why it was chosen in the discussion section Amongst a wide range of polymers that can be used for drug encapsulation, poly(ε-caprolactone) (PCL) has gained increasing interest due to its well-characterized biocompatibility and biodegradability properties and Food and Drug Administration (FDA) approval for a wide range of clinical uses.

We further highlight this aspect in in the discussion section: PCL was chosen as the polymeric carrier due to its biocompatibility, biodegradability, low immunogenicity, and documented sustained drug release in many delivery systems, used in many Food and Drug Administration (FDA)-approved medical devices for a wide range of therapeutic applications [42-44]

Comment:The author needs to compare the present work with other PCL-based microparticles based on the literature”.

Response: We compared our work with another study that used electrospraying to produce PCL particles. This has been included in the discussion section: The average size of the PCL particles ranged from 2 to 10 μm, which is in agreement with previous studies that utilized electrospraying for the fabrication of PCL microspheres [45]. To the best of our knowledge, this was the only study where PCL has been electrosprayed and no studies utilized PCL particles to deliver resveratrol, neither in vitro nor in vivo. For this reason, we compared our system to other polymeric carriers (e.g., PLGA) that have been used for the sustained release of resveratrol or other electrospraying-based systems.

Comment:Please check for grammar corrections”.

Response: The manuscript has been revised and grammar mistakes have been corrected.

Reviewer 4 Report

This is an interesting study describing the preservation of RSV activity for wound healing applications through microparticle formulation. The following comments should be addressed:

The vendors of all chemical should be included in the materials section e.g. line 92 for PCL, line 93 for RSV etc.

2.4. Dynamic light scattering (DLS) analysis: what is the particle concentration dispersed in the buffer?

2.11. In vitro release profile analysis: how did authors assure that in vitro release studies were performed under sink conditions?

What is the rationale for choosing 100 μg/mL as the microparticle concentration to be tested in the cell studies and why the authors didn’t test an array of concentrations? How does the concentration 1 mM of pure RSV selected relate to the respective amount of loaded RSV in the microparticles?

line 299: remove the underlying from the temperature ºC throughout the text

Figure 1D: Check the scale bar used for 10 % RSV-PCL particles since it does not justify the claim that no statistically significant differences were observed in the radii between the different particles 0%-10% RSV-PCL particles. 10 % RSV-PCL particles seem to be significantly smaller.

line 377: it is common that electrospraying/spinning result in drug amorphization, which is not the case for RSV in the microparticles. How would authors justify the presence of RSV in crystalline form in the microparticles?

line 394: superscript -1 in cm-1

line 410: what about the antioxidant activity of the RSV-PCL microparticles beyond 72 h? in vitro release studies showed a sustained release of RSV over 28 days. Does RSV retain its antioxidant capacity over that period of time? Similar for its photodegradation

3.3. Cell attachment, viability, metabolic activity, and proliferation analyses: Provide a detailed description of the experimental findings.

how do authors justify the fact that 10 % RSV-PCL particles induced the highest collagen type I deposition, yet no significant difference was observed in the gene expression of collagen type I, compared to the control.

Figure 9 should serve as introductory figure/graphical abstract

Figure S1: provide the UV scan of the PCL solution alone

Author Response

Reviewer No 4:

Comment:The vendors of all chemicals should be included in the materials section e.g. line 92 for PCL, line 93 for RSV etc.”.

Response: The manuscript has been revised and vendors of all chemicals have been included. Furthermore, in paragraph 2.1 Materials, we clarified that all chemicals and reagents were purchased from Sigma Aldrich (USA), unless otherwise stated

Comment:2.4. Dynamic light scattering (DLS) analysis: what is the particle concentration dispersed in the buffer?”

Response: The particle concentration in the buffer was 2 mg/mL. The following has been added in the Materials and methods section to address this comment: “The samples were prepared in phosphate-buffered saline (PBS) at a concentration of 2 mg/mL and sonicated before the analysis to ensure a dispersed solution”.

Comment: “In vitro release profile analysis: how did authors assure that in vitro release studies were performed under sink conditions?”

Response: At regular intervals, sample aliquots were withdrawn and replaced with the same volume of fresh PBS to maintain the sink conditions. Furthermore, the solubility of RSV in PBS is 100 µg/mL (https://pubchem.ncbi.nlm.nih.gov/compound/Resveratrol), much higher than the average concentrations (1-10 µg/mL) measured during the release studies. We are aware that a paddle apparatus is recommended to perform dissolution tests according to international guidelines, but unfortunately, this was not available for our experiments.

Comment:What is the rationale for choosing 100 μg/mL as the microparticle concentration to be tested in the cell studies and why the authors didn’t test an array of concentrations? How does the concentration 1 mM of pure RSV selected relate to the respective amount of loaded RSV in the microparticles?”

Response: We realized there was a mistake in the manuscript. 1 mM of pure RSV was the concentration of our stock solution. The final concentration that was used for cell culture was 5 µM. We have corrected the manuscript accordingly. 5 µM concentration was chosen based on a literature search. Several studies showed that RSV has a positive effect on cell behavior at a concentration ranging between 1-10 µM (https://pubmed.ncbi.nlm.nih.gov/31068817/; https://pubmed.ncbi.nlm.nih.gov/29935438/), whereas at higher concentration (e.g., 50 µM) RSV could induce cell apoptosis (https://pubmed.ncbi.nlm.nih.gov/27539371/; https://pubmed.ncbi.nlm.nih.gov/31847250/). The rationale for choosing 100 μg/mL as the microparticle concentration was based on preliminary gene expression experiments which showed a reduction of pro-inflammatory markers after stimulation of human dermal fibroblasts with TNF-α. This concentration does not correlate to 5 µM pure RSV, which served only as the positive control. Furthermore, we also tested a higher dosage of particles (500 µg/mL), which caused severe cell detachment, probably due to steric hindrance (see image below), and therefore were excluded from further analysis.

Comment: “Line 299: remove the underlying from the temperature ºC throughout the text”

Response: The symbols have been corrected: Default amplification conditions were as follows: 95 °C for 10 min; 40 cycles of 95 °C for 1 s and 60 °C for 20 s

Comment:Figure 1D: Check the scale bar used for 10 % RSV-PCL particles since it does not justify the claim that no statistically significant differences were observed in the radii between the different particles 0%-10% RSV-PCL particles. 10 % RSV-PCL particles seem to be significantly smaller.”

Response: Initially, we did not statistically compare the radii measured from SEM images. We now performed statistical analysis and found that 10 % RSV-PCL particles were significantly (p < 0.01) smaller in comparison to the other groups. The manuscript has been revised accordingly: PCL (Figure 1E), 2.5 % RSV-PCL (Figure 1F), and 5 % RSV-PCL (Figure 1G) samples were comprised of particles with a diameter range from 5 µm to 10 µm, while the 10 % RSV-PCL (Figure 1H) samples were comprised of particles with a diameter range from 2 µm to 4 µm and were significantly (p < 0.01) smaller in comparison to the other groups. However, no statistically significant (p > 0.05) differences were observed in hydrodynamic radius among all microparticles

Nevertheless, measurements of hydrodynamic radius did not show any statistically significant difference between groups.

Comment:line 377: it is common that electrospraying/spinning result in drug amorphization, which is not the case for RSV in the microparticles. How would authors justify the presence of RSV in crystalline form in the microparticles?

Response: We really appreciate this comment as it helped us explain contradictory results we observed with DSC. Most of our experiments did not show an endothermic peak for RSV at 260 °C in all RSV-PCL microparticles and only a couple of times did we see a peak at 260 °C. The manuscripts as been revised as follows: Differential scanning calorimetry (DSC) analysis (Figure 2B) of control RSV powder revealed a sharp endothermic peak at 260 °C, corresponding to the melting temperature of its crystalline form. DSC of control PCL, 0 %, 2.5 %, 5 %, and 10 % RSV-PCL particles, showed an endothermic peak at 50-70 °C, corresponds to the melting temperature of the crystalline portions of PCL. In the majority of tests, 2.5 %, 5 %, and 10 % RSV-PCL particles DSC thermograms did not show an endothermic peak at 260 °C, indicating that RSV became fully amorphous after the electrospraying process. Only in rare cases, negligible amounts of crystalline RSV could be detected at 260 °C (Supplementary Figure S3A)

Figure 2: Physicochemical analysis of RSV-PCL electrosprayed microparticles. FT-IR analysis (A) revealed the presence of characteristic bands at 3250 cm-1 corresponding to O-H stretching of the RSV, thus confirming its presence in the PCL microparticles (n = 3 batches). DSC analysis (B) of the majority of RSV-loaded microparticle batches showed no endothermic peak at 260 °C that is characteristics for crystalline RSV indicating that RSV became fully amorphous after electrospraying. However, NMR analysis of (C) 2.5 % RSV-PCL, (D) 5 % RSV-PCL, and (E) 10 % RSV-PCL revealed 6 narrow signals in the low field region, above 6 ppm, typical of RSV spectra (n = 3 batches).

Supplementary Figure S3: DSC analysis (A) showed that only in rare cases, negligible amounts of crystalline RSV could be detected at 260 °C. Thermogravimetric (TGA) analysis (B) showed a shift of RSV weight loss upon incorporation in PCL indicating that the polymer enhanced the thermal stability of RSV (n = 3 batches). NMR spectrum (C) of control RSV powder

Comment:line 394: superscript -1 in cm-1.”.

Response:FT-IR analysis (A) revealed the presence of characteristic bands at 3250 cm-1”

Comment:what about the antioxidant activity of the RSV-PCL microparticles beyond 72 h? in vitro release studies showed a sustained release of RSV over 28 days. Does RSV retain its antioxidant capacity over that period of time? Similar for its photodegradation.”.

Response: For the evaluation of the antioxidant capacity, we used a commercially available DPPH Antioxidant Assay Kit (Dojindo Molecular Technologies, USA). Considering that the solvent used to dissolve the DPPH reagent was ethanol, our experiments were limited by solvent evaporation over time and could not get results beyond 72 hours. Future studies could focus on the optimization of assays to measure antioxidant capacity over longer periods of time. Regarding photodegradation, our system could not sustain UV irradiation for a long time. Nevertheless, all the experiments were performed in the dark to minimize the potential of photodegradation of RSV-loaded microparticles.

Comment:Cell attachment, viability, metabolic activity, and proliferation analyses: Provide a detailed description of the experimental findings.”.

Response: This section has been revised: Cell attachment analysis (Figure 4A) revealed that 0 % RSV-PCL and 10 % RSV-PCL microparticles did not negatively affect human dermal fibroblasts attachment as a function of time. Viability analysis (Figure 4B) through Live/Dead assay showed that 0 % RSV-PCL and 10 % RSV-PCL did not significantly alter human dermal fibroblasts' viability as a function of time. Metabolic activity analysis (Figure 4C) through alamarBlue® assay showed no toxicity of 0 % RSV-PCL and 10 % RSV-PCL on human dermal fibroblasts as a function of time. Proliferation analysis (Figure 4D) through the counting of Hoechst-stained nuclei revealed that 0 % RSV-PCL and 10 % RSV-PCL induced an increase in the number of human dermal fibroblasts as a function of time, similar to untreated control cells.

Comment:how do authors justify the fact that 10 % RSV-PCL particles induced the highest collagen type I deposition, yet no significant difference was observed in the gene expression of collagen type I, compared to the control.”.

Response: We believe that by decreasing the expression of MMPs, RSV reduces the degradation of collagen type I over time, leading to the accumulation of ECM.

Comment:Figure 9 should serve as introductory figure/graphical abstract.”.

Response: While we appreciate the comment, we believe that the illustration summarizes concepts and results that were described in the manuscript, pointing out that RSV-PCL may constitute an efficacious wound therapy by interfering with the vicious cycle of inflammation, proteolysis, and inadequate repair and regeneration in chronic wounds. We hence believe Figure 9 is better placed at the end of the manuscript. However, we fully agree that a simplified version of Figure 9 could be very suitable as a graphical abstract, which was added in the revised version.

Comment:provide the UV scan of the PCL solution alone.”.

Response: A UV scan of the PCL solution has been provided in the supplementary information:

Supplementary Figure S1: The wavelength scan for a solution containing 1 mg/mL RSV dissolved in DMSO (A) showed a maximum absorbance wavelength of 330 nm. The wavelength scan for a solution containing 1 mg/mL 10% RSV-PCL microparticles dissolved in DMSO (B) showed that PCL in the solution did not affect RSV’s absorbance spectrum. The wavelength scan for a solution containing 1 mg/mL pure PCL dissolved in DMSO (C).

Round 2

Reviewer 1 Report

The revised mauscript should be accepted and published in the journal.

Author Response

We would like to express our gratitude to the reviewers for their time and constructive criticism. Based on their comments, we have accordingly revised the manuscript

Reviewer 4 Report

Accept in present form.

Author Response

(The authors gave the same response as above.)
